# Pointer Meter Reading Method Based on YOLOv8 and Improved LinkNet

**DOI:** 10.3390/s24165288

**Published:** 2024-08-15

**Authors:** Xiaohu Lu, Shisong Zhu, Bibo Lu

**Affiliations:** School of Computer Science and Technology, Henan Polytechnic University, Jiaozuo 454003, China; 212209020038@home.hpu.edu.cn (X.L.); zss@hpu.edu.cn (S.Z.)

**Keywords:** pointer meter, LinkNet, partial convolution, CBAM, AFF, rotation correction

## Abstract

In order to improve the reading efficiency of pointer meter, this paper proposes a reading method based on LinkNet. Firstly, the meter dial area is detected using YOLOv8. Subsequently, the detected images are fed into the improved LinkNet segmentation network. In this network, we replace traditional convolution with partial convolution, which reduces the number of model parameters while ensuring accuracy is not affected. Remove one pair of encoding and decoding modules to further compress the model size. In the feature fusion part of the model, the CBAM (Convolutional Block Attention Module) attention module is added and the direct summing operation is replaced by the AFF (Attention Feature Fusion) module, which enhances the feature extraction capability of the model for the segmented target. In the subsequent rotation correction section, this paper effectively addresses the issue of inaccurate prediction by CNN networks for axisymmetric images within the 0–360° range, by dividing the rotation angle prediction into classification and regression steps. It ensures that the final reading part receives the correct angle of image input, thereby improving the accuracy of the overall reading algorithm. The final experimental results indicate that our proposed reading method has a mean absolute error of 0.20 and a frame rate of 15.

## 1. Introduction

Pointer meters are widely used in power systems, chemical industry and other traditional industrial fields because of their simple structure, stable operation, strong anti-interference ability, low cost and easy maintenance. However, its readings rely on manual operation, which is relatively inefficient, especially in situations that demand quick monitoring and response. In addition, pointer meters do not have automatic recording and storage functions, and recording data manually is prone to errors. Capturing an image with a camera and using an intelligent algorithm to read the meters in the image can effectively solve the above problem without significantly increasing the cost. Therefore, it is of great practical significance to study a fast, accurate and reliable algorithm for pointer meter reading.

The pointer meter reading algorithm can be broadly divided into three main steps: dial detection, meter component extraction, and reading recognition. Traditional reading algorithms usually adopt template matching [1,2,3] and Hough circle detection [4,5] in the dial detection part. Hough straight line detection [6,7] is a common method in the meter component extraction part. The final reading is usually done by the distance or angle methods. The problem of traditional readout algorithms is that the overall process is cumbersome and inflexible for the template matching algorithm, which can only detect specific meter targets. The limitation of the Hough transform algorithm is that it has poor anti-interference ability, often fail to obtain the desired target, and usually need an extra adjustment.

In recent years, with the development of deep learning and intelligent industry, many researchers have been proposed deep learning-based solutions for pointer meter reading [8,9,10,11,12,13,14,15]. Deng et al. [8] used YOLOv5 to detect the dial. And employed MobileNetv2 as the feature extraction network for Deeplabv3+, this effectively reduced the model size while ensuring accurate segmentation of the scale lines and pointers. In the reading method part, the circular scale is unfolded into a rectangle, obtaining the relative positions of the scale and pointer. Finally, the reading is obtained through distance scaling. Hou et al. [9] used YOLOX to localize the dial in the image, and then used the attention-based U-Net model to segment the meter components. The authors proposed the ECA module to solve the problem of information loss in the original model during the process of replicating, which successfully improves the segmentation accuracy of the model.This provides a more accurate component information for the subsequent reading process. Dai et al. [10] used YOLOv3 to extract the dial in the scene, then combined $U^{2}\mathrm{Net}$ and the spatial attention module to make the model pay more attention to the pointer and scale area to improve the segmentation effect of the model. Finally, the authors utilized the angular method to obtain the final reading results. The innovations of Zhou et al. [11] and others were aimed to turn the segmentation of the whole pointer into the segmentation of the half-pointer to improve the accuracy and speed of the segmentation step. Using OCR to recognize the digits on the dial and utilizing the local angle method in the reading section improved the accuracy of the meter reading in inclined conditions. Zhang et al. [12] used YOLOv7 to detect the dial, segment the pointer using DeepLabv3+, and processed the segmented pointer using refinement algorithms to obtain more accurate results. Finally, they used the angular method to complete the reading. Huo et al. [13] first used the SIFT feature-matching algorithm to adjust the captured tilted dial to a symmetrical upright shape, and then used the improved UNet++ network to segment the scale and pointer regions. Finally, the distance scale method was used for the reading part. Chen et al. [14] first used YOLOv5 to localize the dial in the scene, and then used a modified $U^{2}\mathrm{Net}$ network based on depth-separable convolution and a focal loss function to segment the pointers and scales in the dial. Then, they utilized a polar coordinate transformation algorithm and distance scaling to obtain the final readings. Peng et al. [15] used YOLOv4 to detect the location and category of the meter, after which an improved U-Net image segmentation technique was used to effectively extract the pointers in the region.

In summary, most of the pointer meter reading methods based on deep learning proposed so far use models with a large number of parameters, such as UNet and UNet++, for the segmentation of pointers and scales. It is not only slow in recognition, but also not easy to be deployed on embedded or computationally limited devices. In order to solve the problem, this paper proposes a pointer meter reading method based on an improved LinkNet [16]. Using a lighter semantic segmentation model to segment pointers and scales can accelerate the overall reading process and reduce deployment pressure. The innovations and main contributions of this paper can be summarized as follows:(1)In order to solve the problem of slow recognition speed in pointer meter reading methods, we adopted the faster LinkNet semantic segmentation model and improved it. Specifically, we have reduced the model’s parameters and computational load by employing partial convolutions and reducing the number of encoding–decoding blocks, which makes it easier to deploy. Meanwhile, attention modules and feature fusion modules were introduced to ensure the accuracy of the segmentation.(2)In the subsequent reading process, we propose a rotation correction network based on ResNet18. This network is capable of correcting the rotation of segmented ruler and pointer images to meet the angle requirements for polar coordinate transformation.(3)Extensive experimental validation has been carried out in environments with different angles and perspectives to demonstrate the feasibility of the proposed method.

## 2. Methods

The pointer meter reading method proposed in this paper consists of the following steps: dial detection, semantic segmentation, rotation correction, polar coordinate transformation, and reading calculation. The overall process is shown in Figure 1. The detection of dial in images is to eliminate interference from complex backgrounds, focusing subsequent steps primarily on the meters themselves to improve the accuracy of readings. Semantic segmentation is the most important part of the reading method in this paper, which obtains the positions of pointers and scales and provides key information for subsequent processes. Rotation correction is the correction of the angle of the segmented image, which ensures the integrity of the scale in the rectangular image after polar coordinate transformation, thereby improving the accuracy of the reading. The reading calculation section first expands the circular scale into a rectangle. Then, it obtains the relative position of the scale and pointer based on the horizontal distribution of pixels of different categories. Finally, the final reading is calculated based on the relative position.

### 2.1. YOLOv8-Based Dial Detection

Before recognizing meter readings, it is necessary to locate the dial in the scene. YOLOv8 is one of the more popular target detection algorithms in the industry, and is also the latest target detection algorithm introduced by the YOLOv5 development team. Compared to YOLOv5, YOLOv8 has a higher precision and faster speed. YOLOv8 is divided into five models by setting different network depths and widths: n, s, m, l, and x. YOLOv8n has the smallest parameters and the fastest inference speed among them, so it is also the model selected in this article. Its network structure mainly consists of three parts: backbone, neck, and head. The backbone is used for the extraction of image features at different scales, the neck is used for the fusion of features at different scales, and the head for the prediction of the result output.

The backbone adopts the overall structure of CSPDarkNet, mainly consisting of Conv, C2f, and SPPF modules. The Conv module is used to extract features from the image and contains a convolutional layer, a BatchNormal layer, and an activation function. The C2f module has adopted the ideas from the ELAN in YOLOv7,enabling it to achieve a richer gradient flow of information while maintaining lightweightness. The SPPF module is capable of converting feature maps of arbitrary size into feature vectors of fixed size.

The neck is responsible for fusing multi-scale features and generating a feature pyramid, including the SPPF module and PAN module. The PAN module is designed with a complex bidirectional path to fuse features. It not only adopts a bottom–up path to progressively integrate low-level feature details, but also transmits high-level global contextual information through a top–down path. Through this bidirectional feature fusion mechanism, the PAN module can balance local details and global semantic information, thereby generating a more informative feature representation.

The head section uses a decoupled head structure to separate the classification and detection processes. For the allocation of positive and negative samples, YOLOv8 adopts the Task-Aligned Assigner strategy. This strategy combines classification and regression scores and selects positive samples by weighting these scores, which helps to improve the model’s adaptability to complex scenarios and detection accuracy. For loss computation, the classification branch uses the BCE loss (Binary Cross Entropy Loss), and the detection box regression branch combines DF Loss (Distribution focal loss) and CIOU Loss (complete intersection over union loss) to improve the accuracy of the model’s prediction of the bounding box.

### 2.2. Segmentation of the Meter Pointer and Scale Based on Improved LinkNet

In the currently proposed reading method, the model used for segmenting pointers and scales is relatively large in size. We used a lightweight real-time semantic segmentation model LinkNet and improved it to address these issues. This improvement aims to reduce the complexity and resource consumption of the model while maintaining efficient segmentation performance in environments with limited computing power.

LinkNet adopts an encoding-and-decoding network structure, where the encoding part contains an initialization layer and four coding layers. The initialization layer uses convolution with a step size of 2 and maximum pooling to downsample the input image twice, reducing the image size of the input network, which can effectively improve the operation efficiency of the network. The coding layer uses the network architecture of ResNet18, which consists of a residual structure that effectively extracts key features from the image. The decoding part consists of four decoding layers and a classification layer. The main function of the decoding layer is to recover the size of the image by Transposed Convolution and the classification layer outputs the final segmentation result by classifying the feature map pixel by pixel. In order to make the model more lightweight and easy to deploy, this paper improves it in the following four aspects, and the improved network structure is shown in Figure 2.

Firstly, to reduce redundancy in the model’s computational process, Partial Convolutions (PConv) [17] is employed to replace standard 3 × 3 convolution within the encoder’s residual blocks. This significantly reduces the number of parameters and computations while maintaining accuracy. Secondly, removing a set of encoder and decoder blocks with 512 channels further reduces the number of parameters and computations. This enhances the model’s overall efficiency, albeit at the cost of a slight decrease in accuracy. To compensate for this loss in accuracy, the Convolutional Block Attention Module (CBAM) [18] is introduced in the skip connections, increasing the model’s focus on segmentation targets. Finally, the original direct addition operation for feature fusion is replaced with an Attention Feature Fusion (AFF) [19] module. This module uses two different scale branches to extract global and local features from the features to be fused, then concatenates them with the original feature map. This enables the model to focus on information at different scales, thereby improving the effectiveness of feature fusion.

#### 2.2.1. PConv Module

During traditional CNN (convolutional neural network) training, the generated feature maps tend to show a high degree of similarity between channels, leading to a significant redundancy of information. This situation is not conducive to computational efficiency and may affect the performance of the model with limited resources. To address this problem, PConv significantly reduces redundancy in the computational process by performing convolution operations only on specific major feature channels while keeping other channels unchanged. This approach not only optimizes the computational efficiency, but also reduces the computational burden on the model without sacrificing accuracy.

In the strategies such as DWConv (depthwise separable convolution) and GhostConv (ghost convolution), the parameters and FLOPs (floating point of operations) are often reduced by considering the redundancy of filters. However, to compensate for the reduced precision, such strategies usually choose to increase the number of channels, which leads to an increase in memory access, greatly reducing the computational efficiency. In contrast, PConv efficiently generates the final feature expression by performing partial channel convolution calculations on the feature map and directly adding the results to the remaining channels. As shown in Figure 3, the smaller convolution kernel used for convolution in this process means that the computation is less expensive and faster.
(1)$$FLOPs=2\times H\times W\times K_{h}\times K_{w}\times C_{i}\times C_{o}$$

PConv has smaller FLOPs compared to conventional convolution operations. Assuming the convolution operation uses kernels with the same length and width. Based on the FLOPs Equation (Equation 1), it can be concluded that if the the number of channels $\left(C_{i}\right)$ processed in the PConv is only half of the original number of channels, the FLOPs will also be reduced by half. This significant reduction not only reduces the computational cost, but also increases the processing speed, making PConv particularly suitable for application scenarios with limited computational resources.

#### 2.2.2. CBAM Module

In the feature fusion part of LinkNet, the original LinkNet network directly adds the feature maps’ output from each layer of the encoder to the corresponding decoder outputs. However, this approach does not fully utilize the extracted feature information. Based on this, our paper introduces the CBAM (Convolutional Block Attention Module) and AFF (Adaptive Feature Fusion) module to enhance the effect of feature fusion.

CBAM calculates attention matrices in both channel and spatial dimensions and then multiplies these attention matrices with the input feature maps for adaptive feature learning, the specific structure is shown in Figure 4. If the input feature map is denoted as $F\in R^{CxHxW}$, and the channel and spatial feature extraction modules are represented as $M_{c}$ and $M_{s}$, respectively, then the overall feature extraction process can be described as Equation (Equation 2):(2)$$F^{\prime }=M_{s}\left(M_{c}\left(F\right)⊗F\right)⊗M_{c}\left(F\right)⊗F$$The symbol ⊗ represents element-wise multiplication, and $F^{\prime }$ denotes the final output of the CBAM module.

As shown in Figure 3, the input feature map is first subjected to feature extraction of the channel, the features obtained from maximum pooling and average pooling are fed into the Shared Multilayer Perceptron to learn the weight parameters, and the parameters of the two pooling branches are summed up, and then the learned parameters are normalized by the sigmoid function to obtain the channel attention matrix, This process can be described as Equation (Equation 3):(3)$$M_{c}\left(F^{\prime }\right)=\sigma \left(\mathrm{MLP}\left(\mathrm{AvgPool}\left(F^{\prime }\right)\right)+\mathrm{MLP}\left(\mathrm{MaxPool}\left(F^{\prime }\right)\right)\right)$$
where σ denotes the sigmoid activation function. The obtained attention matrix is multiplied by the original feature map, and the resulting product is fed into the spatial attention module.

The spatial attention module is used to learn the spatial location information in the image, which complements the channel attention. This module, for the input feature map, first performs the operations of maximum pooling and average pooling, then feeds it into a 7 × 7 convolutional layer for feature learning of location information, and finally uses sigmoid to obtain the attention weight coefficients, which can be described as Equation (Equation 4):(4)$$M_{s}\left(F^{\prime }\right)=\sigma \left(f^{7\times 7}\left(\left[\mathrm{AvgPool}\left(F^{\prime }\right);\mathrm{MaxPool}\left(F^{\prime }\right)\right]\right)\right)$$

Finally, the attention weight matrix is then multiplied with the input feature map from this module to obtain the final feature map.

#### 2.2.3. AFF Module

In the current multi-scale feature fusion methods, the most commonly used is direct summation or splicing along the channel, but this is not necessarily the best choice because there may be complex relationships between different feature maps, and simple summation or splicing does not capture the feature interaction information well. Especially when performing feature fusion with long connectivity, it is often the case that low-level features and high-level features are fused. The low-level features contain more positional and detailed information because they have undergone fewer convolutional layers, while the high-level features tend to have stronger semantic information and less detailed information. Directly fusing them through the operation of summation often fails to achieve the effect of efficiently utilizing the information of both features.

AFF is a channel-attention-based feature fusion method with the core structure of MS-CAM; the structure is shown in Figure 5. The structure is capable of aggregating multi-scale contexts along the channel dimensions and is divided into two branches. The left side is used to extract the global context G(X) and the right side is used to extract the local context L(X), which allows us to emphasize both large objects with a more global distribution and small objects with a more local distribution. Given input $X\in R^{C\times H\times W}$, G(X) and L(X) can be expressed by Equations (Equation 5) and (Equation 6) as
(5)$$L\left(X\right)=B(PWConv_{2}\left(\delta \left(B\left(PWConv_{1}\left(X\right)\right)\right)\right)$$
(6)$$G\left(X\right)=B(PWConv_{3}\left(\delta \left(B\left(PWConv_{4}\left(GAP\left(X\right)\right)\right)\right)\right)$$
where *B* stands for BatchNormal, δ stands for Relu, and GAP stands for Global Affirmative Pooling.

To keep it as light as possible, the local context is added to the global context in the attention model while using point-by-point convolution (PWConv) instead of differently sized convolution kernels as a context aggregator. It is notable that L(X) has the same shape as the input features, preserving and highlighting minor details in the lower-level features. The input $X^{\prime }$ of the module can be obtained from Equation (Equation 7):(7)$$X^{\prime }=X⊗M\left(X\right)=X⊗\sigma \left(L\left(X\right)\right)⊕\sigma \left(G\left(X\right)\right)$$
where σ stands for sigmoid activation function, ⊗ stands for element-by-element multiplication, and ⊕ stands for broadcast addition.

In the scene of feature fusion, given two feature mappings *X* and *Y*, and assuming *Y* is the feature mapping with a larger receptive field, then the MS-CAM-based AFF working mechanism can be described as Equation (Equation 8); the structure is shown in Figure 6.
(8)Z=M(X⊕Y)⊗X+(1−M(X⊕Y))⊗Y
where the fusion feature M(X+Y) consists of real numbers between 0 and 1, And the dashed line indicates 1−M(X⊕Y), so that the network, when faced with two features to be fused, instead of treating them equally heavy, performs soft selection or weighted averaging to determine their respective weights through training.

### 2.3. Rotation Correction

Rotation correction is an important part of the reading method in this paper. When the meter is rotated counterclockwise by a certain angle or is in an inverted state, the pattern formed by the transformation of annular scale through polar coordinates is discontinuous, which is not conducive to subsequent readings. Therefore, it is necessary to correct its rotation angle. In order to adapt to different dials to ensure the generalization of the correction method, this paper uses a deep learning method to predict the rotation angle of the segmented image. Compared with the direct correction of the original image, the segmented image does not have other complex dial information, and it is easier for the model to learn the key semantic features to improve the accuracy of the prediction. At the same time, it is easier to use the segmented image to produce a variety of rotational angle datasets

After experimental testing, the CNN network does not regress well to predict the angle of rotation of the dial image after segmentation from 0 to 360°. The reason for this is that the dial image can be largely regarded as an axisymmetric figure. And after a segmented image is symmetrical, the results differ greatly when the features are roughly similar, leading to the fact that the convolutional neural network does not deal with this situation well. To solve this problem, a new branch of classification is added to classify clockwise and counterclockwise rotations of the image, while the regression branch only needs to deal with the 0–180° case. Considering the need to balance the accuracy of the model and its lightweight, ResNet18 is used as the feature extraction part to design the rotation correction network. At the same time, the idea of migration learning is integrated to accelerate the convergence of the model. The network structure is shown in Figure 7.

Through experiments, it is shown that the network has 99% accuracy in classifying the direction of rotation, and the average absolute error of the regression on the rotation angle is 1.2°. The task of 0–360° angle regression is decomposed into rotation direction classification and 0–180° angle regression, which can solve the problem of poor performance of axisymmetric graphs in the regression prediction of rotation angles in a CNN network well.

### 2.4. Reading Methods

The reading method used in this article is the distance-proportional method, which requires prior knowledge of the index value of the meter to be inspected. The detailed process is shown in Figure 8.

Firstly, the rotationally corrected dial is corrupted to make the boundary between the segmented scales more obvious. Then, the annular scale is expanded into a rectangle using polar coordinate transformation. Afterwards, the two-dimensional scale is compressed into a one-dimensional sequence using a projection-like method, and then the sequence needs to be binarized, which can eliminate some small pixel errors caused by segmentation or image deformation, and also obtain a clearer distribution of the scale. Then, traverse the binary sequence and take the center of each pixel block to represent its position in the one-dimensional sequence. This provides the relative positions of all scales and pointers, which can be used to calculate the final reading in the next step.

When calculating the final reading, first determine the number of scale lines to the left of the pointer, denoted as *N*. Calculate the relative position of the pointer within the current scale interval, denoted as *M*. Using I to represent the dial’s division value, the final reading *R* can be expressed as Equation (Equation 9):(9)R=(N+M)×I

## 3. Experimental Section

### 3.1. Experimental Configuration

Our experimental environment is an Ubuntu system with Intel Xeon Silver 4214R @ 2.40 GHz CPU, 128 GB RAM, and a graphics card GeForce RTX 4080. The model is trained with an initialized learning rate set to 0.001, a batchsize of 8, and 200 training rounds. The experimental dataset was self-collected on the Internet and consisted of 920 images under different meter types, backgrounds, lighting, and distances. Part of the data are shown in Figure 9. Each of these columns represents a type of meter; they have different shapes and scale values. Each row shows a different scene.

The above 920 images are expanded to 1420 by random brightness and random contrast changes for the training and validation of the object detection model, and the detection results of the model are used to produce a semantic segmentation dataset. A total of 500 images for semantic segmentation were obtained using the above methods. Through data augmentation strategies, a total of 900 dashboard images were finally obtained for the training and validation of a semantic segmentation model. The data enhancement strategies used include random horizontal flip, random rotation, and generating a black mask with eight random positions and a size of 24 × 8. The data enhancement strategy is shown in Figure 10.

### 3.2. Indicators for the Assessment of the Model

The reading method in this paper requires the segmentation model to segment the scale and pointer as completely as possible. To evaluate the segmentation effect of the model, two indexes, mPA (mean pixel accuracy) and mIoU (mean intersection over union), are chosen in this paper, where mPA represents the mean of the number of pixels predicted correctly for all categories as a proportion of the number of all pixels in that category, which can be expressed mathematically as Equation (Equation 10); and mIoU represents the mean value of the ratio of the intersection and concatenation of the predicted and true values in all categories, which can be expressed mathematically as Equation (Equation 11).
(10)$$mPA=\frac{1}{k+1}\frac{P_{ii}}{\sum_{i=0}^{k}\sum_{j=0}^{k}P_{ij}}$$
(11)$$mIoU=\frac{1}{k+1}\sum_{i=0}^{k}\frac{P_{ii}}{\sum_{j=0}^{k}P_{ij}+\sum_{j=0}^{k}P_{ji}-P_{ii}}$$
where *k* + 1 represents the total number of categories, $P_{ii}$ represents the number of correctly predicted pixels, $P_{ij}$ represents the number of pixels with real category *i* but predicted as *j*, and $P_{ji}$ represents the number of pixels with real category *j* but predicted as *i*. To evaluate the effect of model lightweighting, this paper also chooses three indexes, namely Paras (parameters), FLOPs, and FPS (frames per second) to more comprehensively show the overall performance of the improved model.

### 3.3. Ablation Experiments

To verify the effectiveness of the improved structure proposed in this paper, six sets of ablation experiments are designed in a targeted manner. The experimental results are shown in Table 1, where PC means PConv and REDB means “remove a set of encoding decoding blocks”. Experiment 1 is to replace the 3 × 3 standard convolution in the encoder with PConv, where p represents the ratio of the number of channels participating in the convolution operation to the number of all channels, and when p is 4, it represents that only 1/4 of the channels participate in the convolution operation. Experiment 2 is to remove a set of encoding–decoding blocks on the basis of Experiment 1. Experiments 3 and 4 are to add the CBAM and AFF module, respectively, on the basis of Experiment 2 to verify its effectiveness. Experiments 5, 6, and 7 are to compare the effects of different ratios of the number of channels involved in the convolutional operation in PConv on the improved overall model.

It can be seen from Experiment 1 that PConv is able to significantly reduce the number of parameters and computation of the model without affecting the accuracy. Comparing Experiment 2 and Experiment 3, it can be seen that CBAM is able to make up for the loss of accuracy brought about by the reduction in the encoding–decoding block. This is because the module not only weights the segmented target in the channel dimension, but also enhances the focus on the target region in the spatial dimension. Thus, the overall feature extraction capability of the model is improved. Comparing Experiment 2 and Experiment 4, it can be seen that compared with the direct summing feature fusion method, AFF can obtain more adequate multi-scale contextual information, thus helping to improve the performance of the model. Comparing Experiments 5, 6 and 7 show that the smaller the proportion of the number of channels involved in the convolution operation, the lower the accuracy of the model. However, the proportion of the number of channels involved in the convolutional operation is not linearly related to the accuracy of the model, and it is better to choose 1/4 of the channels involved in the operation than 1/2 of the channels. This further proves that the convolutional neural network has a large redundancy in the operation. PConv can well eliminate these redundancies by extracting some channels to participate in the convolutional operation. Thus, the model can reduce the number of parameters and the amount of computation under the premise of guaranteeing the performance.

### 3.4. Comparative Experiments

To further evaluate the performance of the improved model, the classical semantic segmentation models U-Net, SegNet, DeepLab V3+ as well as the real-time semantic segmentation networks BiSeNet, LETNet, ELANet, ENet are selected for comparison experiments, and the experimental results obtained in the test set are shown in Table 2. By comparing the FPS metrics in the results, it can be seen that the improved LinkNet has a significant advantage in inference speed compared to other models. The segmentation accuracy is slightly lower compared to U-Net, Segnet, and DeepLab V3+. But the number of parameters and computation is small, the computation speed is fast, and it is more suitable to be deployed on embedded and computationally limited devices.

The segmentation accuracy of the model will affect the accuracy of the overall reading method, while the inference speed will affect the reading efficiency. Compared with BiSeNet, the improved LinkNet model performs better overall though the parameter sizes of LETNet, ELANet, and ENet are relatively small. However, by comparing FPS, it can be seen that they are slightly lacking in inference speed.

### 3.5. Analysis of Reading Methods

In this section, we test the meter images of different meter types and different rotation angles using the reading method proposed in this paper; the reading visualization process is shown in Figure 11. The experimental results show that our method is tolerant to the meter image perspective deformation. It also corrects the rotation of the meter image and can be helpful for the polar coordinate transformation algorithm to unfold the circular image.

From Figure 11, it can be seen that the first column is the original image. The second column is the dial obtained after object detection algorithm. The third column is a mixture of the semantic segmentation result and the dial. The fourth column shows the segmented image after angle correction. The fifth column shows the rectangular scale graph formed after polar coordinate transformation of the segmentation result, and the standard scale graph generated on this basis. Row (a) shows the ideal case: the meter is facing the camera, and at this time, which is after the reading process, the results can be accurately obtained. Row (b) shows the dial that has been rotated. Rows (c), (d), and (e) show the dial in the case of perspective angle changes.

The final readings are shown in Figure 12. The results of the above experiments show that the rotation correction model is able to accurately predict the rotation angle for the result after semantic segmentation, thus avoiding the discontinuity of the scale after the polar coordinate change. For the dials with slight perspective distortions, although polar coordinate transformation does not obtain a horizontal scale distribution, an accurate relative distribution can still be obtained through binarization processing.

### 3.6. Comparison of Different Reading Methods

This section compares the different reading methods in terms of both accuracy and speed dimensions. The complete reading methods include target detection, semantic segmentation, rotation correction, and reading. Table 3 shows the results of the comparison using the different reading methods. Accuracy was evaluated using the MAE (Mean Absolute Error) metric; speed was evaluated using FPS. Method 1 uses YOLOv5 as a detector and Deeplabv3+ as a component segmentation model to extract some of the main scales. Method 2 utilizes YOLOv5 as the detector and $U^{2}\mathrm{Net}$ as the component segmentation model to extract the full scale. And method 3 uses YOLOX as a detector and U-Net as a segmentation model to extract some of the main scales. The experimental results show that the reading method proposed in this paper has a faster detection speed compared to other methods with similar reading accuracy.

## 4. Conclusions

In this paper, we propose a novel pointer meter reading method using a lighter semantic segmentation model, which can improve overall reading efficiency and reduce deployment pressure. Firstly, the dial in the scene is detected using YOLOv8. After that, the relevant components in the dial are segmented using the improved LinkNet network and the rotation is corrected by the proposed deep learning-based rotation correction model. Finally, the accurate readings of the meter are obtained using the distance scaling method.

Experiments have been conducted to evaluate the feasibility and robustness of the presented method. The experimental results show that the segmentation model accuracy of the improved LinkNet proposed in this paper is 88.43% and the fps is up to 247. Combined with the proposed rotation correction method, the overall reading method has an average absolute error of 0.20 and an fps of 15. The proposed algorithm has a certain degree of tolerance for images with slight perspective distortions.

Although in this paper we have reduced the overall reading time by replacing the segmentation model, this time is not enough for real-time performance. And we have not tested a large number of real scenes, in which factors such as bright light, reflection, and dirt may affect the accuracy of the reading algorithm. In addition, experiments have also shown that the inference speed of the model has a large impact on the overall reading process time. Therefore, our next research direction is to start from the lightweight of the detection model. At the same time, optimize the intermediate steps of the reading process in order to achieve the real-time requirements of the overall reading speed.

## Figures and Tables

**Figure 1 sensors-24-05288-f001:**
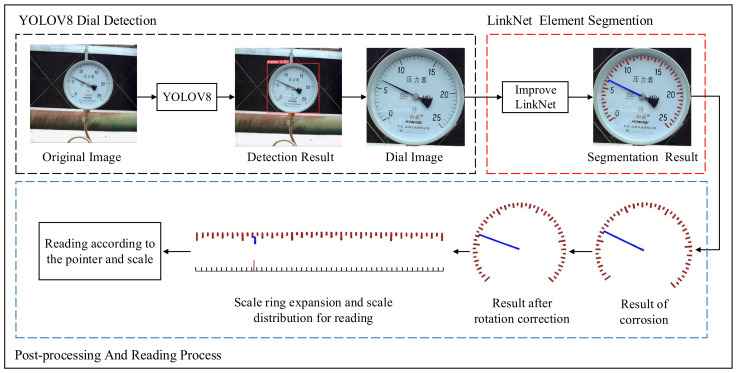
The principle of the pointer meter reading recognition method.

**Figure 2 sensors-24-05288-f002:**
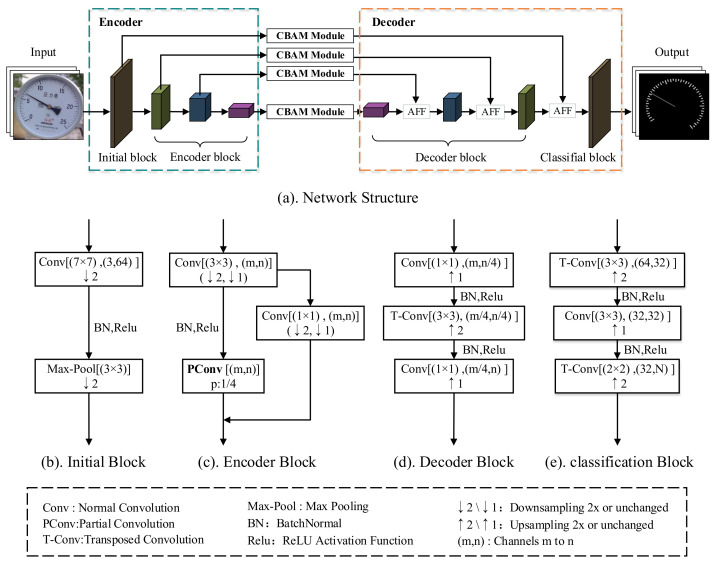
Improved LinkNet architecture.

**Figure 3 sensors-24-05288-f003:**
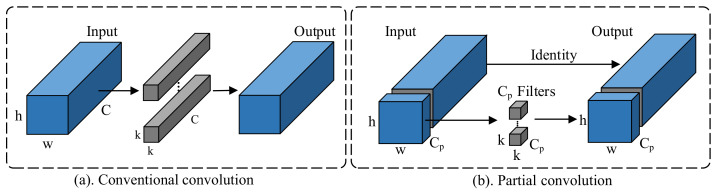
Contrast between conventional and partial convolution.

**Figure 4 sensors-24-05288-f004:**
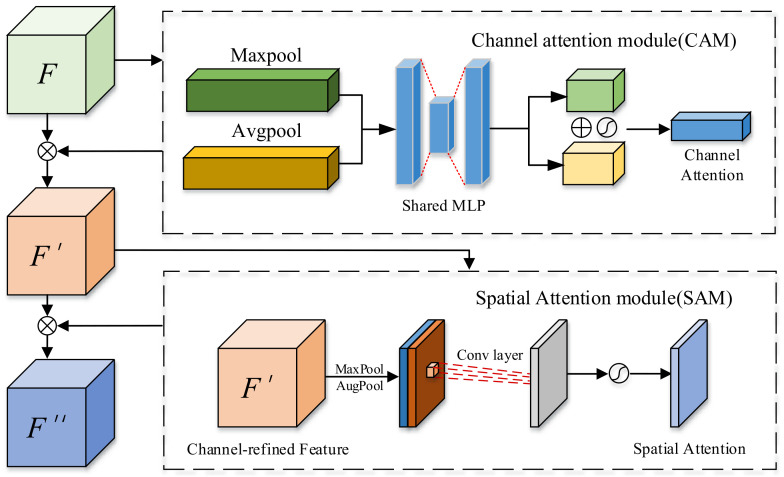
Schematic diagram of the CBAM mechanism.

**Figure 5 sensors-24-05288-f005:**
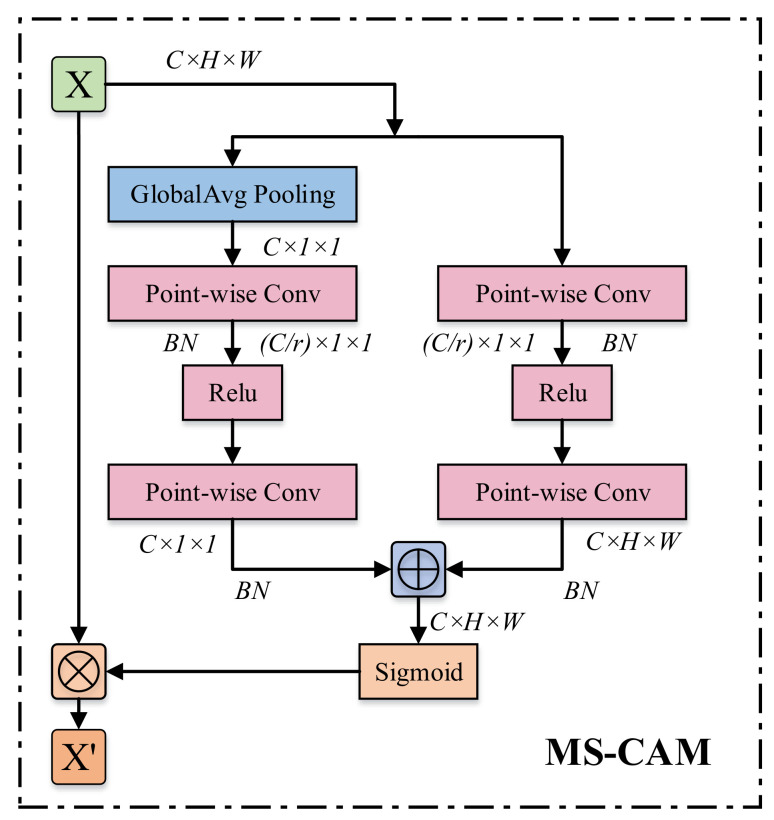
MS-CAM structure diagram.

**Figure 6 sensors-24-05288-f006:**
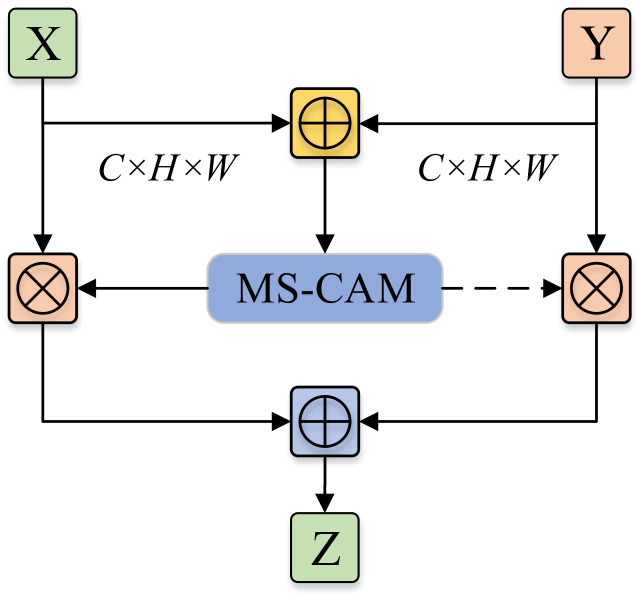
AFF structure diagram.

**Figure 7 sensors-24-05288-f007:**
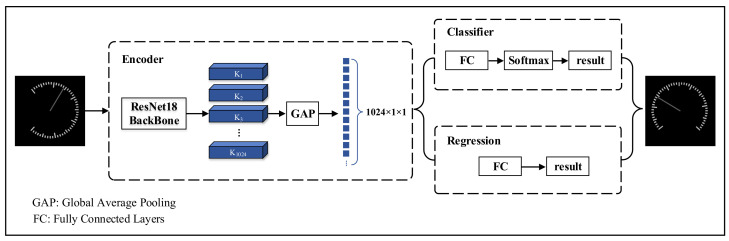
Rotation correction model structure diagram.

**Figure 8 sensors-24-05288-f008:**
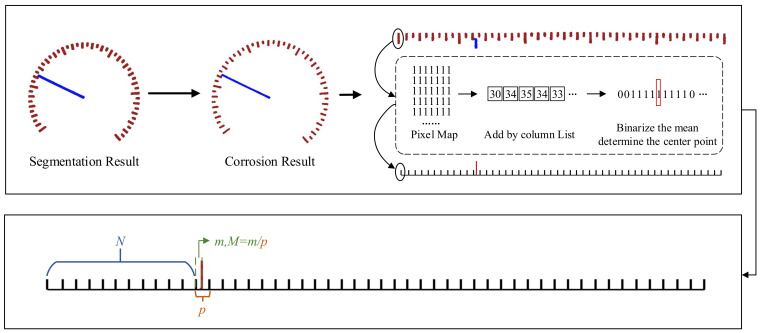
Reading detail display.

**Figure 9 sensors-24-05288-f009:**
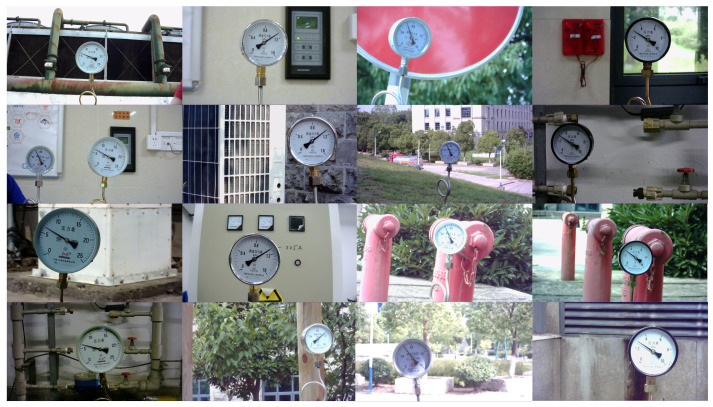
Partial dataset display diagram.

**Figure 10 sensors-24-05288-f010:**
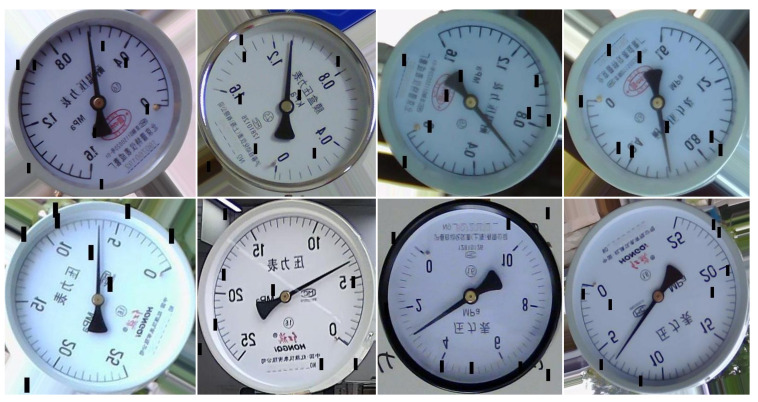
Data Enhancement strategy schematic.

**Figure 11 sensors-24-05288-f011:**
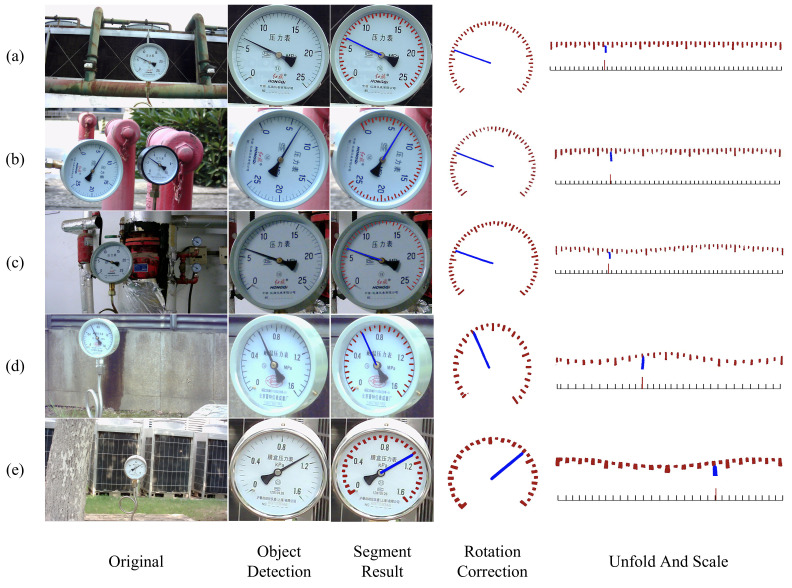
Reading visualization process.

**Figure 12 sensors-24-05288-f012:**
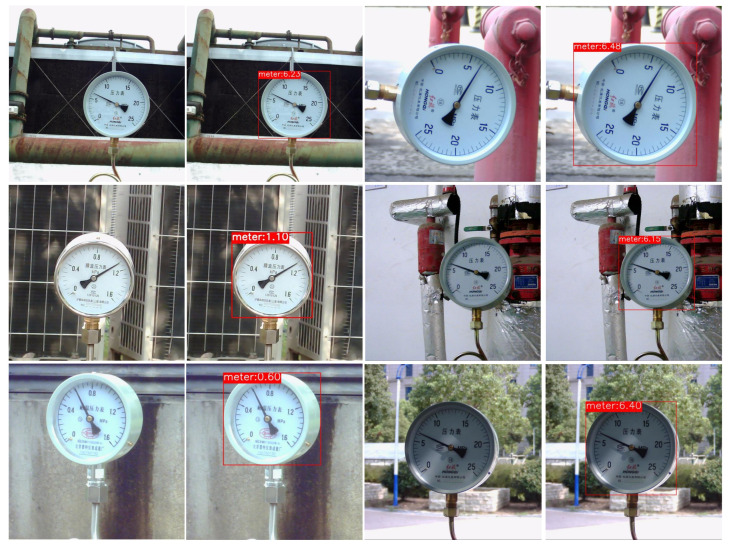
Reading result.

**Table 1 sensors-24-05288-t001:** Ablation experiment.

Model	PC (p = 8)	PC (p = 4)	PC (p = 2)	CBAM	AFF	REDB	mIoU/%	Paras/M	FLOPs/G	FPS
LinkNet							87.86	11.54	12.14	231
1		✓					87.72	6.35	8.15	256
2		✓				✓	86.86	1.61	6.80	304
3		✓		✓		✓	87.83	1.65	6.80	280
4		✓			✓	✓	87.64	1.63	6.91	268
5	✓			✓	✓	✓	87.96	1.60	6.75	258
6			✓	✓	✓	✓	88.35	1.97	7.60	243
7		✓		✓	✓	✓	88.43	1.68	6.92	247

**Table 2 sensors-24-05288-t002:** Comparative experiments of different semantic segmentation models.

Model	mIou	mPA/%	Paras/M	FLOPs/G	FPS
U-Net [20]	90.30	94.73	24.89	225.85	61
SegNet [21]	88.79	94.17	29.46	321.65	88
DeepLab V3+ [22]	88.85	94.35	5.81	52.87	176
BiSeNet [23]	87.86	94.23	23.08	40.7	197
LETNet [24]	88.94	93.29	0.95	13.6	156
ELANet [25]	87.39	93.11	0.67	9.8	100
ENet [26]	86.74	93.46	0.37	0.22	105
Improved LinkNet	88.43	94.12	1.67	6.92	247

**Table 3 sensors-24-05288-t003:** Comparative experiments of different reading methods.

Method			MAE	FPS
Index	Detect	Segment		
Method 1 [9]	YOLOX	U-Net	0.20	9
Method 2 [14]	YOLOv5	$U^{2}\mathrm{Net}$	0.23	10
Method 3 [8]	YOLOv5	Deeplab V3+	0.21	12
Ours	YOLOv8	Improved LinkNet	0.20	15

## Data Availability

The datasets presented in this study can be found in online repositories. The names of the repository/repositories and accession number(s) can be found below: https://drive.google.com/drive/folders/1CoM5Rm5GBg61HrrMG1–hZEX1RBYxWk7?usp=drive_link (accessed on 1 August 2024).

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
