# Peer review of "Pointer Meter Reading Method Based on YOLOv8 and Improved LinkNet"

_sensors, 2024, doi:10.3390/s24165288_

Round 1
Reviewer 1 Report
Comments and Suggestions for Authors
I believe that this manuscript has made significant progress in the research on the reading method of pointer meters, particularly in the segmentation of meter components using an improved LinkNet network. The LinkNet network has undergone various improvements, such as the adoption of partial convolutions and the introduction of attention modules, which serve to reduce model parameters and computational costs while enhancing performance. These advancements can provide valuable solutions for related fields.However, the manuscript suffers from the following issues:
- 
The abstract is relatively complex and could be more concise and straightforward in conveying the core ideas. While the experimental description of the dataset is mentioned, a detailed description of the dataset is lacking, such as its specific composition and sample diversity. 
- 
The self-constructed dataset lacks clarity. To demonstrate the objectivity and enhance the credibility of the experimental results, it is recommended that the dataset be made publicly available. 
- 
The experimental comparison methods are relatively outdated. It is suggested to incorporate more recent comparison methods to validate the effectiveness of the model. Additionally, it seems that ENet exhibits advantages; clarification on this point would be helpful. 
- 
In the conclusion section, in addition to mentioning future research directions, a more thorough summary of the limitations of this study and its implications and assistance for subsequent research could be provided. 
- 
The English proficiency demonstrated in the paper needs to be further improved. 
- 
The English proficiency demonstrated in the paper needs to be further improved. 
- There are also some issues with the typesetting. Please double-check the formatting of the paper carefully.
Reviewer 2 Report
Comments and Suggestions for Authors
1. The innovation point of this article mainly describes the author's contribution, and the description of experimental results should not appear in the innovation point.
2. The comparative experimental results in Table 2 do not reflect the recognition speed advantage of the algorithm proposed by the author.
3. Why are the comparison criteria for Comparative Experiments in Tables 2 and 3 inconsistent? The algorithm frame rate proposed by the author in Table 3 is also higher than other algorithms, which conflicts with the problem description in the Introduction.
4. Suggest the author to check some of the English descriptions in the article.
Comments on the Quality of English Language1. The innovation point of this article mainly describes the author's contribution, and the description of experimental results should not appear in the innovation point.
2. The comparative experimental results in Table 2 do not reflect the recognition speed advantage of the algorithm proposed by the author.
3. Why are the comparison criteria for Comparative Experiments in Tables 2 and 3 inconsistent? The algorithm frame rate proposed by the author in Table 3 is also higher than other algorithms, which conflicts with the problem description in the Introduction.
4. Suggest the author to check some of the English descriptions in the article.
Round 2
Reviewer 1 Report
Comments and Suggestions for Authors
Why add those two comparison methods? Need to explain the reasons and provide a brief introduction.
Comments on the Quality of English LanguageEnglish expression needs further enhancement.
Reviewer 2 Report
Comments and Suggestions for Authors
The revised manuscript of the paper has been modified in response to relevant opinions.
Comments on the Quality of English LanguageIt is recommended that the author carefully review the relevant descriptions during proofreading.
